# Quasi-localized vibrational modes, Boson peak and sound attenuation in model mass-spring networks

Shivam Mahajan[1] and Massimo Pica Ciamarra[1,2,3]⋆

**1** Division of Physics and Applied Physics, School of Physical and Mathematical Sciences, Nanyang Technological University, Singapore
**2** CNRS@CREATE LTD, 1 Create Way, #08-01 CREATE Tower, Singapore 138602
**3** CNR–SPIN, Dipartimento di Scienze Fisiche, Università di Napoli Federico II, I-80126, Napoli, Italy
⋆ massimo@ntu.edu.sg

November 3, 2022

## Abstract

We develop an algorithm to construct mass-spring networks differing in the correlation length of their local elastic properties. These networks reproduce Boson peak and quasi-localized vibrational modes observed in glasses and their relationship. The sound attenuation rate also behaves as in glasses, varying from a Rayleigh scattering to a disorder-broadening regime on increasing the frequency. We find that networks modelling glasses with increased stability have a reduced elastic correlation length. Our results support a deep connection between the various vibrational features of disordered solids and clarify their relationship with the correlation length of the local elastic properties.

# 1  Introduction

The density of vibrational states (vDOS) controls solids' specific heat and transport properties [1,2]. The vDOS of amorphous solids differs qualitatively from that of crystals. In crystals, the low frequency vibrational excitations are plane waves (phonons) distributed in frequency according to Debye's law, $D_p(\omega) \propto \omega^{d-1}$ in $d$ spatial dimensions. On the contrary, amorphous materials have an excess of low-frequency modes over Debye's prediction that induces a peak in the reduced density of states $D(\omega)/\omega^{d-1}$ at the Boson peak frequency, $\omega_{\rm bp}$, in the terahertz regime for molecular solids. Previous works have attributed the Boson peak to elastic disorder [3–5], localized harmonic/anharmonic vibrations [6–9], broadening of van Hove singularities [10,11] (but see [12]). In addition, in amorphous solids the low-frequency excitations comprises both phononic-like modes and additional quasi-localized vibrational modes (QLMs) that appears to be universally distributed in frequency as $D_{\rm loc}(\omega) = A_4 \omega^4$ [13–16], with $A_4$ decreasing as the stability of the material increases [17]. The low-frequency $\omega^4$ scaling of the density of soft-localised modes may originate from general considerations on the properties of localised excitations in disordered systems [18] (see, however [19]). The universality of the scaling law [14,15] and its dimensionality independence suggest that this scaling has a mean-field origin. Yet, mean-field replica theory recovers this scaling only upon incorporating finite dimensional fluctuations [20]. Finally, in amorphous materials the extended low-frequency modes are not phonons: even in the absence of temperature induced anharmonic effects [21], phonons of wave vector $\kappa$ attenuate with a rate $\Gamma(\kappa)$ exhibiting a crossover from a Rayleigh scattering [22] regime, $\Gamma \propto \kappa^{d+1}$, to a disordered-broadening regime, $\Gamma \propto \kappa^2$ [23–27], as $\kappa$ increases.

The squared vibrational eigenfrequencies $\omega^2$ are the eigenvalues of the matrix of the second derivatives of the energy with respect to the particle positions or Hessian matrix. As such, the vibrational anomalies of amorphous materials may possibly be rationalized within random matrices [28–33]. Previous works primarily focused on the eigenvalues of Wishart matrices, which are positively defined and hence may model stable systems. A mean field [34] random-matrix approach suggests that the Boson peak may originate from the reduction in coordination number driving the system toward isostaticity [29, 35] and from hierarchical energy landscape. These two scenarios are possibly relevant in colloidal hard-sphere-like glasses and highly connected molecular systems. The random matrix approach may also be used to investigate QLMs. In this case, the issue is determining the random matrix ensemble reproducing the $D_{\rm loc}(\omega)$ distribution characterizing amorphous solids or, equivalently, the correlations to be enforced on the matrix. Research in this direction [36] succeeded in reproducing a pseudogap, $D(\omega) \propto \omega^\alpha$, with an exponent $\alpha < 4$. In this random-matrix research direction, the issue is integrating the two approaches to reproduce at the same time Boson peak and quasi-localized modes, as well as their correlations.

Other approaches recovered the $\omega^4$ distribution by describing an amorphous material as an elastic continuum punctuated by defects, possibly anharmonic or interacting [37–40]. Localized vibrations may thus cause all vibrational anomalies of glasses, considering that they may induce the Boson peak [6–9] and control sound attenuation in Rayleigh's theory [22]. Recent numerical results supported this scenario by demonstrating a relation between QLMs' frequency distribution and Boson peak frequency [41,42], $A_4 \propto \omega_{\rm bp}^{-5}$, and by showing that vibrations with frequencies close to the Boson peak consist of phonons hybridized with QLMs [43].

In this manuscript, we investigate the physical origin of the vibrational anomalies of amorphous solids by creating mass-spring networks, or equivalently Hessian matrices, that reproduce them and their relationships. Rather than looking for the ensemble of random matrices exhibiting the anomalies of interest, we study how to vary an amorphous solid's mass-spring network to modulate them. Similar approaches have been introduced to induce a Boson peak

in unstressed networks [44] or suppress $A_4$ by artificially reducing the prestress in stressed ones [45]. Here, we introduce an algorithm to decrease the spatial correlation length $\xi_e$ of the local shear modulus of stressed networks without sensibly affecting the prestress. We show that both $\omega_{bp}$ and $A_4$ decrease with $\xi_e$ and that they satisfy the relation $A_4 \propto \omega_{bp}^{-5}$ observed in model stable glasses. Our networks also reproduce sound's attenuation crossover from a Rayleigh scattering to a disordered broadening regime observed in amorphous solids. In the Rayleigh scattering regime, the attenuation rate relates to the material properties as predicted by fluctuating elasticity theory [3,4], which assumes no correlations in the elastic properties, despite our networks having correlated elastic properties. Our results support a deep connection between the vibrational anomalies and glasses and clarify their relationship with the spatial correlation of the local elastic properties.

The paper is organised as follows. We introduce our approach to construct mass-spring networks in Sec. 2, and verify in Sec. 4 that our methodology allows tuning the correlation of the elastic disorder. In Sec. 5, we demonstrate that the vibrational spectrum of our networks reproduces the vibrational anomalies of amorphous materials and their correlations [42]. Sec. 6 shows that sound attenuation in our synthetic elastic networks behaves as in amorphous materials. Finally, Sec. 6 shows that sound attenuation in our networks crossovers from Rayleigh scattering regime $\Gamma \sim \omega^3$ to disorder-broadening regime $\Gamma \sim \omega^2$, as expected for two-dimensional solids. We summarise our results and discuss future research directions in the conclusions.

## 2 Numerical model and protocols

Our mass-spring network generating algorithm takes as input the disordered mass-spring network associated with the linear response regime of an amorphous solid, which generally has bond-depending elastic constants and rest lengths. We transform this original network by swapping the attributes of randomly selected bond pairs, i.e., by exchanging their spring constants and rest lengths, as schematically illustrated in Fig. 1. We define the fraction of swapped bonds as $f = 2N_{swap}/N_b$, where $N_{swap}$ is the number of swap moves, $N_b$ the number of bonds in the network, and the factor 2 accounts for the fact that each swapping event involves two bonds. Hence, for $f = 0$ we retain our original network, while for $f = 1$ each bond has been swapped once on average. After the bond swapping, we minimize the energy of the new network bringing it into a mechanically stable configuration and study its vibrational properties.

We remark that, on increasing $f$, the bond randomization procedure destroys the correlations in the local elastic properties of the initial network more effectively. However, regardless of the $f$ value, the elastic properties of the final network may exhibit correlations that build up during the final minimization procedure. The exact relation between $f$ and correlation in the elastic properties needs to be determined a posteriori.

We have implemented our bond-swapping procedure in two dimensions. To generate our initial network, we consider systems of particles interacting via the Weeks-Chandler Andersen (WCA) potential

$$U_{ij}(r_{ij}) = 4\epsilon \left[ \left( \frac{\sigma_{ij}}{r_{ij}} \right)^{12} - \left( \frac{\sigma_{ij}}{r_{ij}} \right)^6 \right] + \epsilon; \ \ r_{ij} \leq 2^{1/6} r_{ij}, \tag{1}$$

with $r_{ij}$ the distance between interacting particles, $\sigma_{ij} = (\sigma_i + \sigma_j)/2.0$ where $\sigma_i$ is a particle diameter drawn from a uniform random distribution in the range [0.8:1.2]. We equilibrate systems at fixed number density $\rho = 1.2$ at high temperature $T = 4\epsilon$, and then instantaneously quench them into amorphous solid configurations by minimizing the energy via the conjugate-gradient algorithm [46].

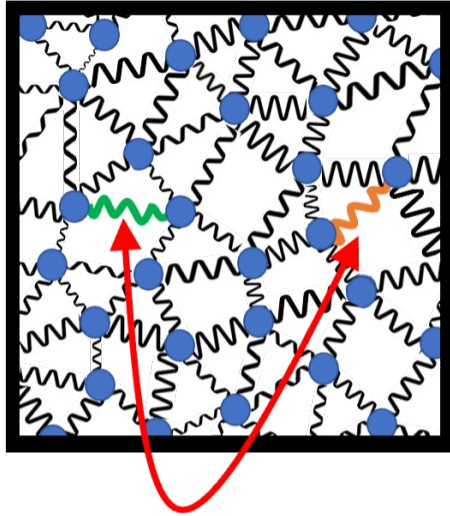

Figure 1: Schematic of the bond-swapping algorithm we use to tune the elastic properties of a disordered mass-spring network. In each bond swapping event, we randomly select two elastic springs and swap the values of their elastic constants and rest lengths. We obtain different network by varying the fraction of swapped bonds $f = 2N_{\text{swap}}/N_b$, where $N_{\text{swap}}$ is the number of swap moves and $N_b$ the number of bonds in the network, and then minimizing the energy of the resulting network.

We generated initial mass-spring networks with $N = 1024$ to $360000$ particles. Each network is fed to our algorithm to create different networks by swapping a fraction $f$ of the bonds. For each $N$ and $f$, we average our data over 200 independent initial networks unless otherwise specified.

## 3 $f$ dependence of mechanical and geometrical properties

We investigate the influence of the bond-swapping procedure on the geometrical and mechanical properties of the elastic network in Fig. 2. Panel a shows that bond swapping does not affect two-point correlations as the radial distribution function is de-facto $f$-independent. In panel b, we study the $f$ dependence of the distribution of the interparticle forces F. We obtain our initial $f = 0$ network by minimising the energy of a system of particles interacting via a repulsive potential. Consequently, for $f = 0$ all interparticle forces are positive, i.e., repulsive. The swapping protocol changes the interparticle forces' distribution by inducing tensile forces. These changes influence the network's mechanical properties by increasing the shear modulus, as in Fig. 2c. These results suggest that bond-swapping leads to networks resembling those associated with very stable glasses.

Our swapping protocol has a minor effect on the prestress of the system, which we evaluate [47,48] as $e = (d-1)\langle(-U'(r_{ij})/r_{ij}U''(r_{ij})\rangle_{ij}$. Fig. 2d shows that the prestress monotonically decreases on increasing the swapping fraction. The prestress change is small, about 5%. A large fraction of this change occurs on moving from $f = 0$ to $f > 0$, i.e., as tensile forces appear in the system. The small prestress variation suggests that our approach to tune the mechanical properties of the network differs from previous ones [45] that directly affect the prestress.

The results of Fig. 2b and d further clarify that the $f$-induced changes in the vibrational properties we discuss in the following do not relate to variation in the connectivity occurring close to the jamming point [49], or analogous, to the emergence of many weak contacts leading

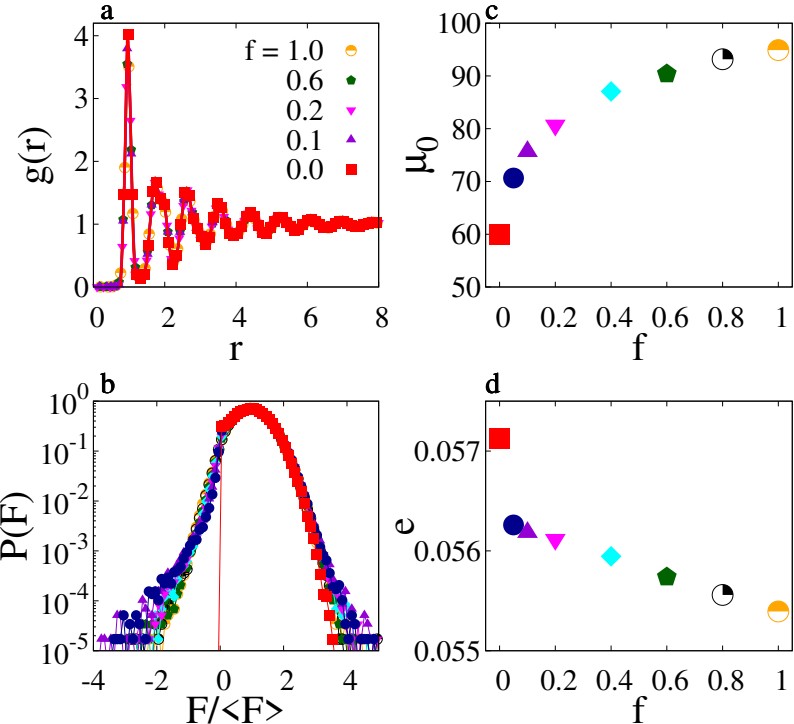

Figure 2: Panel (a) illustrates the radial distribution function of $N = 40000$ particle systems. Panel (b) shows the distribution of the magnitude of the spring forces $F$ normalized by their average value. Data are averaged over 200 configurations. Panel (c) illustrates the dependence of shear modulus $\mu_0$ on the swapping fraction $f$. Panel (d) shows the variation of pre-stress $e$ with $f$. Data (b)-(d) are for $N = 1024$. Different symbols refer to different $f$ values, e.g., as in (b).

to a small effective connectivity [50].

## 4 Disorder Parameter and Fluctuations of Elastic Properties

We quantify how the swapping probability $f$ influences elastic disorder via the dimensionless disorder parameter $\gamma$ introduced in Schirmacher's fluctuating elasticity [3, 51]. If local elastic properties are short-range correlated, the fluctuations $\sigma_w$ of an elastic property coarse-grained over a large length scale $w$ are inversely proportional to the number of particles in that region, $N_w \propto w^d$ in $d$ spatial dimensions. The disorder parameter $\gamma$ fixes the scaling of the normalized shear modulus' fluctuations,

$$\frac{\sigma_w^2}{\mu_0^2} = \frac{\gamma}{N_w}, \tag{2}$$

with $\mu_0$ is the macroscopic shear modulus. As such, $\gamma$ measures the number of close particles with correlated elastic properties, or equivalently, a correlation length $\xi_e \propto \gamma^{1/2}$ in two spatial dimensions. The study [42, 45] of the fluctuations of the elastic properties on the system size $N$, rather than on $N_w$, similarly allows estimating the disorder parameter.

Here, we adopt the local elasticity approach. Henceforth, we define a per-particle stress tensor and investigate its variation under imposed external deformations to evaluate per-particle elastic constants [42]. which we average of square observation window of side length $w$. We provide details in Appendix A. Fig. 3a illustrates the scaled fluctuations of local shear modulus as a function of coarse-graining length scale $w$. These scaled fluctuations approach a

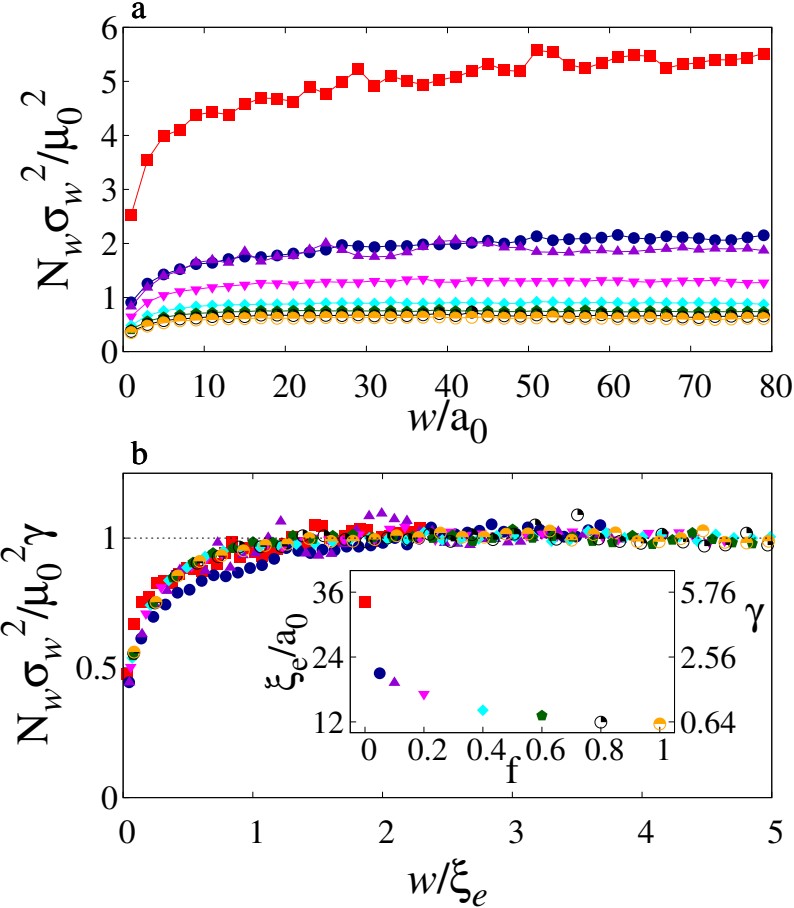

Figure 3: (a) Dependence of fluctuations of the local shear modulus on the coarse-graining length scale $w$. (b) Data of panel (a) collapse when their magnitude is scaled by $\gamma$, and $w$ is scaled by $\xi_e \propto \gamma^{1/2}$. The inset of (b) illustrate the dependence of the elastic length scale $\xi_e$ and disorder parameter $\gamma$ on the swapping fraction $f$.

$f$-dependent asymptotic value with increasing $w$, demonstrating that they are asymptotically governed by the central limit theorem. The asymptotic value thus corresponds to the disorder parameter $\gamma$, which we found to decrease on increasing the swapping fraction $f$.

If $\xi_e \propto \gamma^{1/2}$ is the only length scale influencing the fluctuations of the shear modulus, then these fluctuations must vary with the coarse-graining length scale as

$$\frac{\sigma_\mu^2}{\mu_0^2} = \left(\frac{\xi_e}{w}\right)^2 g\left(\frac{w}{\xi_e}\right) = \frac{\gamma}{N} g\left(\frac{w}{\xi_e}\right), \tag{3}$$

with $g(x)$ a universal function asymptotically approaching. We verify this scaling in Fig. 3(b), and fix the constant of proportionality between $\xi_e$ and $\gamma^{1/2}$ enforcing $g(x)$ to be constant for $x = w/\xi_e \gtrsim 1$.

The inset of Fig. 3(b) shows that the correlation length decreases and approaches a constant value as the fraction of swapped bonds increases. This observation suggests that by increasing the swapping fraction $f$, we suppress correlations and obtain mass-spring networks of glasses with increased stability.

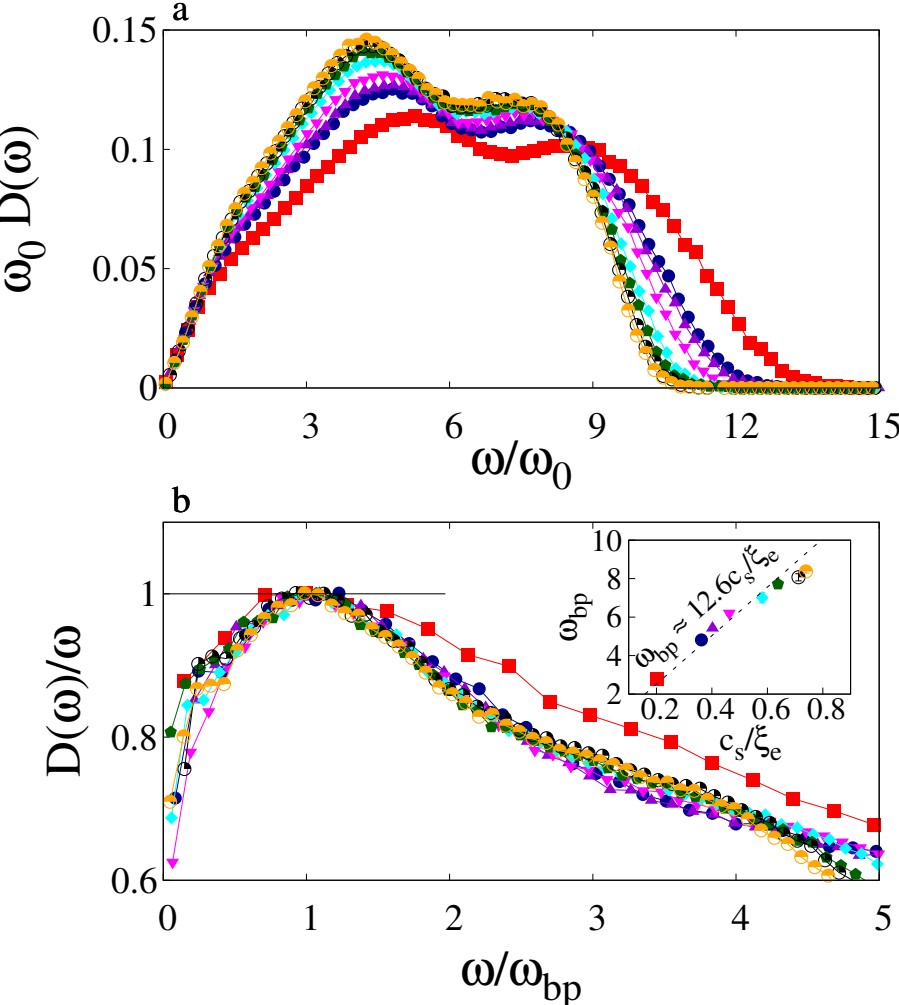

Figure 4: Panel (a) illustrates the density of states of system with $N = 160000$ as a function of $\omega/\omega_0$ for different swapping fractions $f$. Panel (b) shows the reduced $D(\omega)$, normalized by its maximum value, as a function of $\omega/\omega_{\mathrm{bp}}$. In the inset, we show that $\omega_{\mathrm{bp}} \simeq 0.84 c_s/\xi_e$.

## 5 Vibrational Spectra

We now show that the bond-swapping algorithm leads to elastic networks whose vibrational properties exhibit a Boson peak and QLMs and investigate how these vibrational anomalies relate to the elastic length scale $\xi_e$.

### 5.1 Boson Peak

We investigate the $f$ dependence of the Boson peak by determining the vibrational density of states of large $N = 160000$ systems by Fourier transforming the velocity auto-correlation function. Fig. 4(a) illustrates the vDOS for different $f$ values, upon scaling the frequency by $\omega_0 = c_s/a_0$, with $c_s = \sqrt{\mu_0/\rho}$ the shear-wave speed and $a_0 = \rho^{-1/2}$ the interparticle spacing. The reduced vDOS $D(\omega)/\omega$ exhibits a boson peak at characteristic $f$ dependent frequency $\omega_{\mathrm{bp}}$ which we have extracted from the data collapse of Fig. 4(b). The inset shows that the Boson peak frequency scales as $c_s/\xi_e$, suggesting that the excess vibrational modes stem from vibration occurring over the elastic length scale, as observed in model three-dimensional

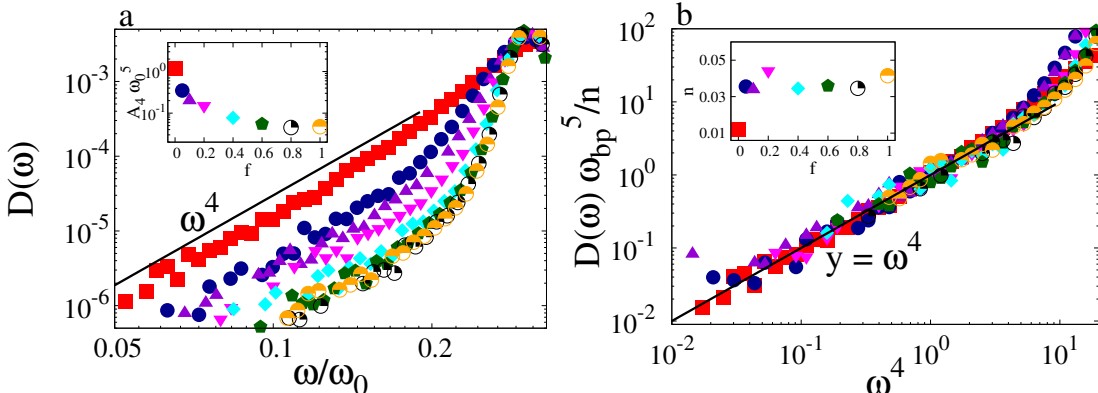

Figure 5: (a) $D_{\text{loc}} = A_4 \omega^4$ scaling of low-frequency density of states. Different symbols indicate different swapping fraction $f$. The inset illustrates the $f$ dependence of the prefactor $A_4$. (b) A plot of $D_{\text{loc}} \omega_{\text{bp}}^5 / n$ vs $\omega$, with $n$ a number density that weakly depends on $f$ as in the inset, leads to a data collapse. For each $f$, data are averaged over $50,000$ configurations of $N = 1024$ particles.

glasses [42].

We remark that according to correlated fluctuating elasticity theory [51] (corr-FET) the Boson peak frequency depends both on the ratio $c_s / \xi_e$ and on the disorder parameter $\gamma$. Yet, the analysis of the solutions of corr-FET equations reported in Ref. [51] indicates that $\omega_{\text{bp}} \sim \frac{c_s}{\xi_e} (\gamma - \gamma_c)^q$, with $q \simeq 0.4$ and $\gamma_c$ the critical value of the disorder parameter at which the system becomes unstable. For highly connected stable molecular glasses as the ones we are considering, $\gamma \ll \gamma_c$, and corr-FET thus predicts $\omega_{\text{bp}} \propto c_s / \xi_e$. These considerations further indicate that increasing the fraction of swapping bonds or equivalently reducing the correlation of the local elastic properties leads to elastic networks resembling those associated with stable glasses.

## 5.2 Quasi-Localized Modes

We investigate the vibrational spectrum' low-frequency end via the direct diagonalization of the Hessian matrix. We focus on small $N = 1024$ systems to shift the lowest phonon frequency ($\omega_{min} \propto c_s / L$) upwards and expose the QLMs, and perform average over 50000 realizations for each $f$ value. For all swapping probabilities, $f$, QLMs are distributed in frequency as $A_4 \omega^4$, as illustrated in Fig. 5a. This result supports the speculated universality of the scaling exponent. The amplitude $A_4$ decreases on increasing $f$, as in the inset, again suggesting that networks with larger $f$ mimic those of glasses with increased stability.

The amplitude $A_4$ has the units of a density of modes over a frequency to the power 5. If $\omega_{\text{bp}}$ is the QLMs' characteristic frequency, then $A_4 = \omega_{\text{bp}}^5 / n$ with $n$ the number density. In Fig. 5 we find that if $D(\omega) \omega_{\text{bp}}^5 / n$ is plotted versus $\omega^4$ data for different $f$ collapse on a master curve, $A_4 \omega_{\text{bp}}^5$ having a week $f$ dependence, particularly for $f > 0$, as in the inset. This result establishes a close correspondence between Boson peak and QLMs. A similar result holds in three-dimensional glasses [41, 42]. In that case, however, the density of modes $n$ resulted smaller by a factor of ten.

While in the system considered here and in two model stable glasses of Refs. [41, 42] $n$ results essentially constant, the universality of this result deserves further investigations and, if validated, a theoretical explanation. In this respect, we note that using as QLM's characteristic frequency one related to the bulk-averaged response of amorphous solids to force dipoles, Ref. [16] found more stable glasses to have a smaller $n$. Hence, either is some systems $n$ vary

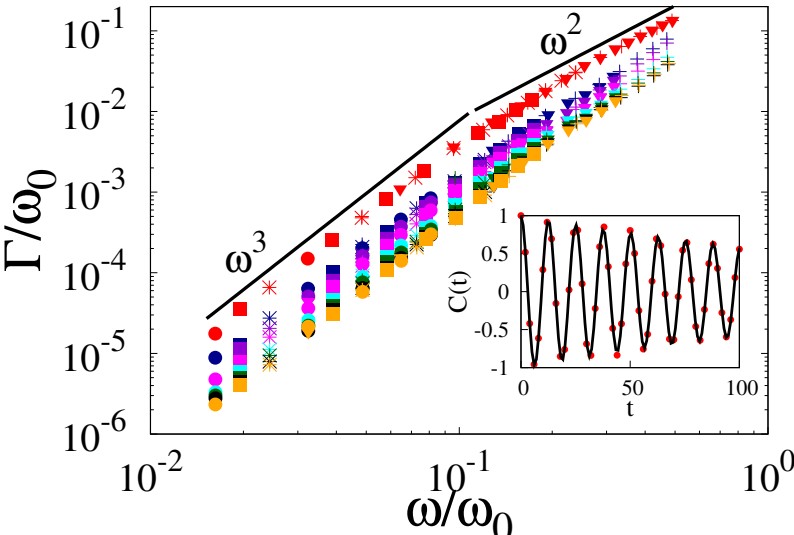

Figure 6: Dependence of the phonons' attenuation rate on the frequency normalized by $\omega_0$. Colors correspond to different swapping fractions $f$, as in Fig. 5b. Different symbols correspond to different system sizes, $N$=40000 (+), 90000 ($\bigtriangledown$), 160000 ($\star$), 250000 ($\square$) and 360000 ($\circ$). The inset shows the velocity auto-correlation function $C(t)$ of transverse phonons of wavevector $k$ excited at $t = 0$, with superimposed damped cosine wave fit, for $N = 160000$, $f = 0.0$, and $\kappa^2 = 9$.

on increasing the degree of supercooling, or the characteristic frequency extracted in Ref. [16] does not scale as the Boson peak frequency.

## 6 Phonon Attenuation

We now discuss how swapping influences phonon attenuation. To evaluate the phonon attenuation rate, $\Gamma$, we excite [52, 53] a transverse acoustic wave by giving each particle a velocity $\mathbf{v}_i^0 = \mathbf{A}_T \cos(\boldsymbol{\kappa r}_i^0)$, where $\mathbf{A_T}\boldsymbol{\kappa} = \mathbf{0}$, considering $\boldsymbol{\kappa}$ in which one among $\kappa_x$ and $\kappa_y$ is zero, and evaluate the velocity auto-correlation function:

$$C(t) = \frac{\sum_{i=1}^{N} \mathbf{v}_i(0).\mathbf{v}_i(t)}{\sum_{i=1}^{N} \mathbf{v}_i(0).\mathbf{v}_i(0)}. \tag{4}$$

We remind we work by definition in the linear response regime as we are considering the response of a system of masses and springs. For each $\kappa = |\boldsymbol{\kappa}|$, we average this correlation function over 30 phonons from independent samples for $N \leq 360000$. Finally, we extract attenuation rate $\Gamma$ and frequency $\omega$ as a function of wave-vector $\kappa$ by fitting the velocity auto-correlation function to a damped oscillation, $\cos(\omega t)e^{-\Gamma t/2}$. As an example of this procedure, we show in the inset of Fig 6 the velocity autocorrelation function for $\kappa = 3$ in a $N = 160000$ system and its damped exponential fit.

Fig 6 illustrates the dependence of the attenuation parameter on $\omega/\omega_0$. At all $f$ values, we observe the crossover from strong Rayleigh scattering $\Gamma \sim \omega^3$ to disorder-broadening regime $\Gamma \sim \omega^2$ with increasing frequency $\omega$ as found in glasses. At fixed $\omega/\omega_0$, the attenuation parameter $\Gamma$ reduces with increasing $f$. We remark that these results do not suffer from size effects, as we explicitly show combining data for different N values.

Rayleigh's original model [22] explains the $\omega^4$ scaling of the attenuation rate by describing the elastic medium as an elastic continuum punctuated by isolated defects. Specifically, it

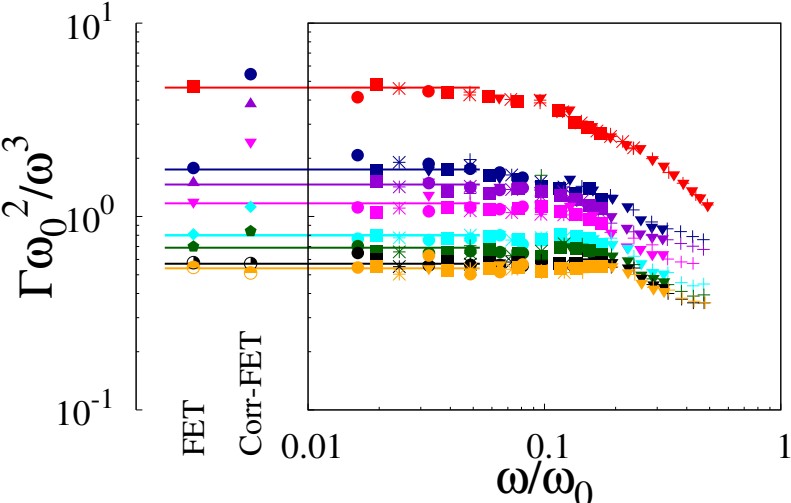

Figure 7: Scaled attenuation rate as a function of the frequency. The asymptotic value in the $\omega \to 0$ limit is compared tp to the predictions of classical FET, which assumes the local elastic properties to be $\delta$-correlated, and corr-FET, which accounts for the local elasticity to be spatially correlated.

relates the attenuation rate to the number density of defects, their size, and the deviation of their shear elastic properties from the background. In terms of the disorder parameter $\gamma$, Rayleigh's prediction results [42] $\Gamma\omega_0^2/\omega^{d+1} = a\gamma^2$ with $a$ a constant, and coincides with corr-FET predictions [51]. We have found Rayleigh's to work in three-dimensional model stable glasses by identifying the scattering defects with the QLMs [42].

We do not expect the same scenario to hold in the systems considered here. Indeed, from the typical QLMs' number density, $n$, and size $\xi_e$, we estimate these occupy an area fraction of order $n\xi_e^2 \simeq 5$. This high value implies that the assumption of isolated defects at the heart of Rayleigh's model does not hold. In Fig. 7 we illustrate the dependence of $\Gamma\omega_0^2/\omega^{d+1}$ on $\omega/\omega_0$, and compare the low frequency plateau with the corr-FET prediction, with the constant $a$ chosen to capture the $f = 1$ data (orange). Corr-FET prediction does not work for the other $f$ values, in agreement with our expectation.

In Fig. 7 we also compare the attenuation rate with FET's prediction, $\Gamma\omega_0^2/\omega^{d+1} = a\gamma$. This prediction is obtained under the assumption that the elastic disorder is $\delta$ correlated [3], or equivalently, that the disorder correlation length is fixed, and has been also recovered by studying the disordered induced broadening of the width of phonon bands [54]. We find that FET well describes our data. Ref. [45] reported a similar result in a two-dimensional model systems. We find puzzling that FET correctly describes sound attenuation in our system and a matter deserving further investigations. Indeed, FET's assumption of $\delta$ correlated elastic properties does not hold in our systems, as $\xi_e$ varies with $f$ as illustrated in the inset of Fig. 3b.

# 7 Conclusions

We have introduced an algorithm that modulates the correlation length of the local elastic properties of a mass-spring network without affecting the network's connectivity and prestress. For all values of the correlation length of the local shear modulus $\xi_e$, the vibrational density of states exhibits a boson peak at a frequency $\omega_{bp}$, which scales as $\omega_{bp} \propto c_s/\xi_e$. The vibrational density of states also features quasi-localized vibrational modes distributed in frequency as $D_{loc} = n\omega^4/\omega_{bp}^5$, with $n$ a number density weakly dependent on the elastic length $\xi_e$. As such,

the generated networks exhibit the typical vibrational anomalies of stable molecular glasses, and these anomalies are related as recently observed in model stable glasses [42].

The generated networks also recover the sound wave attenuation rate's crossover from a low-frequency Rayleigh scattering to a disordered broadening regime characterizing amorphous solids in the harmonic approximation. While the frequency scaling of the attenuation rate scales as predicted by Rayleigh's theory for the scattering of plane waves by isolated defects, $\Gamma = C\omega^3$ in two spatial dimensions, the proportionality factor $C$ does not depend on the local elastic properties as predicted by this theory. We rationalize this result considering that the QLMs cannot be considered isolated defects as their number density $n$ and size $\simeq \xi_e$ are too large. We find the proportionality factor $C$ to depend on material properties as predicted by fluctuating elasticity theory, which assumes uncorrelated local elastic properties. This finding is surprising as the local shear modulus of our networks is spatially correlated and poses a question deserving further investigation.

Our algorithm influences the correlation length of the elastic properties by affecting the elastic correlation length $\xi_e$ while keeping constant the number density $n$ of localized defects. It would be interesting to devise other algorithms to tune $n$, mainly to suppress it, if possible. In this respect, plans include using the algorithm to tune the elastic properties of three-dimensional networks, as in three dimensions glasses [42], $n$ is smaller than the one we reported here by a factor of $\simeq 10$.

Overall, our results support the existence of a strong correlation between QLMs and Boson peak and show that both vibrational anomalies relate to the correlation length of the local shear elasticity.

**Acknowledgments**    We thank E. Lerner and E. Bouchbinder for comments on a earlier version of this manuscript.

**Funding information**    We acknowledge support from the Singapore Ministry of Education through the Academic Research Fund Tier 1 (2019-T1-001-03) and Tier 2 (MOE-T2EP50221-0016) and are grateful to the National Supercomputing Centre (NSCC) of Singapore for providing the computational resources.

# A   Local elasticity

Within two-dimensional linear elasticity the macroscopic stress and the macroscopic strain are related by $\sigma_\alpha = c_{\alpha\beta}\epsilon_\beta$, with $\alpha, \beta \in \{xx, yy, xy\}$ and $c_{\alpha\beta}$ the stiffness tensor. Here, we define the local stiffness matrix $c_{\alpha\beta}^{\text{cg}} = \frac{d\sigma_\alpha^{\text{cg}}}{d\epsilon_\beta}$ as the ratio between a locally defined stress and the macroscopic strain [42,55]. We define the coarse grained stress as $\sigma_\alpha^{\text{cg}}(w) = \langle \sigma_\alpha^{(i)} \rangle$, where the average is over all particles $i$ in the coarse-graining volume, and $\sigma_\alpha^{(i)}$ is a per-particle stress [56].

We note that other approaches could be used to define local elastic properties [57, 58], e.g., by introducing a locally defined stresses and strains. These diverse definitions converge for large coarse-gaining lengths but differ at finite $w$. The definition we have adopted here recover self-averaging. In addition, with this definition the statistics of the elastic properties coarse grained over a sub-region containing $N_0$ particles of a $N \gg N_0$ system match those of a $N_0$ particle system [42].

Practically, we evaluate $c_{\alpha\beta}$ by monitoring the change in the stresses of the particles in response to small deformation followed by energy minimization, to capture the non-affine contribution to the elasticity, making sure we work in the linear response regime. We coarse

grain the single-particle elastic properties over square regions of side length $w$.

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
