# Peer review of "Quasi-localized vibrational modes, Boson peak and sound attenuation in model mass-spring networks"

_SciPost Physics_

## Round 1 · Referee Report · Anonymous (Referee 1) · 2022-12-5

Strengths

In this paper authors proposed a link between soft vibrational modes, Boson peak and sound attenuation in disordered spring networks. The main results are: 1) The development of a swapping algorithm that enables the generation of spring networks that feature a wide range of mechanical disorders. 2) Numerous important mechanical properties are measured and compared including: pre-stress, fluctuations of shear moduli, non-phononic density of states, Boson peak frequency, and wave attenuation rates. 3) A discussion of linking these properties to a glassy length scale (i.e the shear modulus spatial correlation) is proposed.

Weaknesses

1) Generality of the algorithm developed is not discussed. 2) One of the main claim that the minute decrease of pre-stress does not affect shear modulus fluctuations is not correct 3) Lack of connection with previous studies on the connection between a glassy length scale associated with quasi-localized excitations and elastic properties of disordered solids.

Report

I would be happy to recommend this paper as a publication in scipost if the requested changes are addressed.

Requested changes

Major comments: 1) The statement that the proposed algorithm does not affect pre-stress is not entirely true. Yes I agree that in percent it is relatively small (about 3-5%) but parallel works have shown that this amount is already enough to significantly decrease the fluctuations of the shear modulus. For example in [Ref.45 & arXiv:2012.03634], authors have reported a change of \gamma about a factor of 9 (factor 3 in what they defined as \chi with \gamma=\chi^2) for a change of prestress around 5%. This result is fully consistent with the numbers reported in the present paper. I would therefore change the statement that the algorithm does not affect prestress, as they have a deep connection with the decrease of \xi_e and \gamma.

2) I would revise the discussion about the correlation length associated with the shear modulus spatial distribution and its connection with \gamma, BP peak, and wave attenuation rate. It was previously established in the literature that the length scale associated with the core of QLMs \xi_qlm\sim c_s/\omega_qlm, with \omega_qlm a characteristic frequency for localized soft modes [Ref.16] . It was speculated that the same frequency is matching the location of the BP peak, which was recently demonstrated using a non-linear framework [arXiv:2210.10326]. This glassy length was also shown to scale with the disorder parameter \gamma as \xi\sim\gamma^{1/d} [González-López, Karina, et al. PRE 103(2021)]}. The correlation length of the shear modulus distribution is simply the core size of QLMs (there are no other mesoscale length scales in amorphous solids). It explains why \xi_e\sim\gamma^{1/2} and \omega_{BP}\sim 1/\xi_e (as shown in the present paper and in Ref. 41 in 3d). This paper would gain in clarity with a dedicated discussion making the direct link between \xi_e and \xi_qlm.

3) The original network used in this study is constructed from glasses featuring only repulsive interactions at a finite positive pressure. Could authors check if the decrease in \xi_e with the fraction of bond swapping f is also observed in glasses with attractive interactions prepared at zero pressure.

4) Authors should put in perspective their results on the prediction of FET for sound attenuation with recent studies (e.g. Szamel & Flenner, JCP, 156 (2022)) showing that FET largely underestimates wave attenuation rates.

Minor comments: - Citation 17 in the introduction is probably a mistake and should be replaced by citation 41 (same authors) where A_4 is reported for swapped Monte Carlo glasses. - In Equation 2, I would define explicitly for non-experts what is \sigma_W. - page 5 after “to evaluate per-particle elastic constants” there is a full stop instead of a comma. -Change "localised" to "localized" page 2

  • validity: high
  • significance: high
  • originality: good
  • clarity: high
  • formatting: good
  • grammar: good

Author:  Massimo Pica Ciamarra  on 2023-03-21  [id 3501]

(in reply to Report 1 on 2022-12-05)

Dear Reviewer,

Many thanks for your feedback on our manuscript.
We have performed additional studies and modified the manuscript, and provide a detailed point-to-point response to your comments in the attached file.

Sincerely,
the authors

Attachment:

Understanding_the_vibrational_anomalies_in_simple_mass_sprin_dCGoLQ8.pdf

---

## Round 1 · Referee Report · Anonymous (Referee 2) · 2023-1-6

Strengths

This work presents an algorithm which systematically modifies local elasticity in amorphous systems. By doing this, we can systematically and quantitatively examine the relation between local elasticity and vibrational properties, which is one of the major subjects in physics of amorphous solids. The authors basically demonstrate the consistency between their numerical simulation results and the prediction of the fluctuating elasticity theory.

Weaknesses

The algorithm that the authors propose is to swap the bond between particles and break the correlation of local elasticity, which seems able to control the fluctuation (sigma_w) and correlation length (xi_e) of local elastic constants. However, this algorithm is too artificial to my opinion. We usually quench the system to generate amorphous configurations, and the slower quenching generates the more stable system with less fluctuations. I am not sure if the present results are related to real amorphous materials.

Report

Although the method is too artificial, the results seem to be reasonable. The relation between local heterogeneities and vibrational properties is an important issue, and the present works can contribute to this issue. I think the paper could be considered for publication, if the authors could address issues and questions below.

Requested changes

(1)
Fluctuations of elastic properties are central quantities in this work. So, the authors should present distribution of local stiffness. What is the functional form of distribution? The fluctuating theory assumes several different functional forms of distributions (please see PHYSICAL REVIEW B 88, 064203 (2013)). The authors could discuss numerical results with the theoretical assumption.

(2)
If I correctly understand, the correlation length xi_e is measured as the corase-grained length that the disorder parameter converges. But, a more honest way to measure the correlation length is to calculate the spatial correlation function of the local stiffness. The authors could check the correlation function decays with the distance xi_e.

(3)
What is the meaning of the scale of xi_e ~ gamma^1/2?

(4)
This work considers only the shear modulus. How about the bulk modulus? If the bulk modulus is much larger than the shear modulus, it can not be important for low-frequency vibrational properties. The authors should add this point on discussion.

(5)
In fig2c, the shear modulus increases from 60 to 90. I do not yet understand why the modulus is change so much, by the algorithm of swapping. The prestress shows only a tiny change, so I guess change of the connectivity leads to the increase of shear modulus. Is this correct? The authors would put comments on this point with some figure of conncetivity.

(6)
The authors study the boson peak frequency, but it is good to study the boson peak strength as well. The boson peak strength is measured as D(omega_BP)/D_debye(omega_BP) (where D_debye(omega) is the Debye DOS), which is also described by the fluctuating theory. How is the elasticity fluctuation (or disorder parameter gamma) related to the BP strength?

(7)
In Fig6, crossover between omega^3 and omega^2 seems to vanish as f increases (fluctuations become small). Why so? Is this consistent with the theoretical prediction?

  • validity: high
  • significance: good
  • originality: good
  • clarity: high
  • formatting: excellent
  • grammar: excellent

Author:  Massimo Pica Ciamarra  on 2023-03-21  [id 3500]

(in reply to Report 2 on 2023-01-06)

Dear Reviewer,

Many thanks for your comments on our manuscript.
We have performed additional studies and modified the manuscript accordingly.
We provide a list of changes and a point-to-point response to your questions in the attached file.

Sincerely,
the authors

Attachment:

Understanding_the_vibrational_anomalies_in_simple_mass_sprin_gBNP2Cc.pdf

---

## Editorial Decision

resubmitted